

# Comparison of the fecal microbiota of two free-ranging Chinese subspecies of the leopard (*Panthera pardus*) using high-throughput sequencing

Siyu Han[1,*], Yu Guan[1,*], Hailong Dou[2], Haitao Yang[1], Meng Yao[1], Jianping Ge[1] and Limin Feng[1]

[1] Northeast Tiger and Leopard Biodiversity National Observation and Research Station, Ministry of Education Key Laboratory for Biodiversity Science and Ecological Engineering, State Forestry and Grassland Administration Key Laboratory for Conservation Ecology of Northeast Tiger and Leopard National Park, State Forestry and Grassland Administration Amur tiger and Amur leopard Monitoring and Research Center, College of Life Science, Beijing Normal University, Beijing, China
[2] College of Life Sciences, Qufu Normal University, Shandong, China
[*] These authors contributed equally to this work.

Corresponding author
Limin Feng, fenglimin@bnu.edu.cn

## ABSTRACT

The analysis of gut microbiota using fecal samples provides a non-invasive approach to understand the complex interactions between host species and their intestinal bacterial community. However, information on gut microbiota for wild endangered carnivores is scarce. The goal of this study was to describe the gut microbiota of two leopard subspecies, the Amur leopard (*Panthera pardus orientalis*) and North Chinese leopard (*Panthera pardus japonensis*). Fecal samples from the Amur leopard ($n = 8$) and North Chinese leopard ($n = 13$) were collected in Northeast Tiger and Leopard National Park and Shanxi Tieqiaoshan Provincial Nature Reserve in China, respectively. The gut microbiota of leopards was analyzed via high-throughput sequencing of the V3–V4 region of bacterial 16S rRNA gene using the Life Ion S5™ XL platform. A total of 1,413,825 clean reads representing 4,203 operational taxonomic units (OTUs) were detected. For Amur leopard samples, *Firmicutes* (78.4%) was the dominant phylum, followed by *Proteobacteria* (9.6%) and *Actinobacteria* (7.6%). And for the North Chinese leopard, *Firmicutes* (68.6%), *Actinobacteria* (11.6%) and *Fusobacteria* (6.4%) were the most predominant phyla. *Clostridiales* was the most diverse bacterial order with 37.9% for Amur leopard and 45.7% for North Chinese leopard. Based on the beta-diversity analysis, no significant difference was found in the bacterial community composition between the Amur leopard and North Chinese leopard samples. The current study provides the initial data about the composition and structure of the gut microbiota for wild Amur leopards and North Chinese leopards, and has laid the foundation for further investigations of the health, dietary preferences and physiological regulation of leopards.

## INTRODUCTION

Leopards (*Panthera pardus*) are currently the most widely distributed wild felids (*Jacobson et al., 2016*), but they are confronted with worldwide population declines due to illegal poaching, prey depletion, habitat fragmentation, and anthropogenic disturbances (*Balme, Slotow & Hunter, 2009*; *Hebblewhite et al., 2011*; *Kissui, 2008*; *Nowell & Jackson, 1996*; *Packer et al., 2011*; *Stein et al., 2016*; *Sunquist & Sunquist, 2002*). The International Union for Conservation of Nature (IUCN) recognizes nine subspecies of leopards, including the Amur leopard and the North Chinese leopard (*Miththapala, Seidensticker & O'Brien, 1996*; *Uphyrkina et al., 2001*). The Amur leopard has been classified as critically endangered by IUCN since 1996 (*Jackson & Nowell, 2008*). Once patrolling from Northeast China to southernmost portions of the Russian Far East and the Korean peninsula (*Nowell & Jackson, 1996*), the Amur leopard is currently confined to the adjacent habitats in the Jilin and Heilongjiang provinces in China and southwestern Primorsky Krai in Russia (*Feng et al., 2017*; *Hebblewhite et al., 2011*). The North Chinese leopard originally distributed North and Central China but lost as much as 98% of their historic range. An accurate distribution area and population size still remain unclear due to the lack of empirical investigation (*Jacobson et al., 2016*). Recently, the Cat Classification Task Force of the IUCN Cat Specialist Group revised the taxonomy of leopards and included the North Chinese leopard in Amur leopard on account of the obscure biogeographical barrier between them (*Kitchener et al., 2017*), although North Chinese leopard was described as the typical subspecies in North China since 1862 (*Allen, 1938*; *Gray, 1862*). Moreover, other evidence based on molecular biology supporting this classification for the two leopards are scarce, especially in North China.

Amur Leopard and North Chinese leopard are large-sized feline species and solitary predators that play pivotal roles in the ecosystems where they occur. Many efforts have been made to uncover their dietary habits, population genetic structure, and individual identification for conservation purposes through non-invasive sampling of feces (*Dutta & Seidensticker, 2013*; *Dutta et al., 2012*; *Rodgers & Janečka, 2013*; *Yang et al., 2018*). Gut microbial diversity analyses based on leopard fecal samples should also be considered as an important part of conservation efforts. In-depth understanding of the relationship between host habitat and microbiota composition may be helpful for conservation efforts because changes in the gut bacterial communities have been shown to affect host metabolism and energy homeostasis (*Musso, Gambino & Cassader, 2010*).

The gut microbiota composition is an indicator of health condition for endangered wild animals, since habitat degradation may affect host health negatively via diet-associated shifts in the gut microbiota (*Amato et al., 2013*). Dietary changes caused by human disturbance and habitat degradation likely result in a decrease in microbiota diversity (*Barelli et al., 2015*). Animal groups from habitat under increased anthropogenic pressure could be distinguished by the comparison of gut bacterial communities (*Gomez et al., 2015*). Therefore, changes in the gut microbiota species composition of endangered animals might be used as an indicator of habitat degradation and fragmentation (*Barelli et al., 2015*).

Additionally, it has been shown that the detection of pathogenic bacteria is indicative of severe infectious diseases in endangered species (*Zhao et al., 2017*), and fecal bacterial composition could alter accordingly with gastrointestinal diseases in animals (*Suchodolski et al., 2012*). Research has shown that the fecal bacterial species richness was decreased and various bacterial taxa were altered in cats with diarrhea (*Suchodolski et al., 2015*). Compared with healthy cats, cats with clinical signs of gastrointestinal tract disease had significantly lower amount of microaerophilic bacteria (*Johnston et al., 2001*).

Residential gut bacteria are also able to serve as a natural barrier against invasive pathogens (*Gibson et al., 1995*), and to facilitate the function of the immune system (*Maynard et al., 2012*; *Round & Mazmanian, 2009*). Specific compositions of the gut microbiota are associated with variations in the host diet, phylogeny, and physiological status (*Benson et al., 2010*; *De et al., 2010*; *Nelson et al., 2013*; *Sommer & Bäckhed, 2013*). Characterization of gut bacterial communities is important in understanding the mechanisms of host–microorganism interactions (*Nicholson et al., 2012*). Thus, the gut microbiota analysis is fundamental and of paramount importance to the conservation of endangered species. Potentially, the studies of the gut microbiota could be an assistant tool for understanding the phylogenetic relationship of leopard subspecies in the future.

Here, we characterized and compared the fecal bacterial communities of the Amur leopard and North Chinese leopard via high-throughput sequencing targeting the V3–V4 hypervariable region of the bacterial 16S rRNA gene, and provide the first benchmark of gut bacterial diversity in Amur and North Chinese leopard that potentially contribute to further conservation research.

## MATERIALS AND METHODS

### Sample collection

Opportunistic fecal sampling occurred in the period from December 2016 to March 2017 in the two distribution areas from the leopards. A total of eight (O1-O8) fecal samples of the Amur leopard were obtained from the Northeast Tiger and Leopard National Park located in the Heilongjiang and Jilin provinces of China (E129°05′–131°18′, N42°37′–44°10′). This distribution area of the Amur leopard is characterized by a monsoon climate with cold and windy winters, the main vegetation types are mixed broad-leaved forests and secondary Mongolian oak (*Quercus mongolica Fisch. ex Ledeb*) forest (*Tian et al., 2015*). A total of 13 fecal samples (J1-J13) of the North Chinese leopard were collected from Tieqiaoshan Provincial Nature Reserve in Shanxi province (E111°25′–114°17′, N36°39′–38°06′). This region belongs to the warm temperate continental climate with little snow in winter and dry wind in spring, and the vegetation forms are temperate deciduous broad-leaved forests (*Zheng et al., 2009*). Field experiments were approved by the Forestry Department of Jilin Province, State Forestry Administration and Forestry Department of Shanxi Province.

All fecal samples were collected simultaneously by several groups of our team in different sites. We designed line transects that leopards regularly used based on camera trapping data and sent trained members in field soon after snowfall to collect feces above snow layer where leopard footprint traces were present. Each line transect has been revisited more

than once at a three day interval, only newly-excreted feces after previous inspection were collected with the sterile tools. The low environmental temperature below 0 °C contributed to the preservation of gut microbes in the fecal samples. Samples were stored in special ice boxes under −20 °C during in-field study and finally stored under −80 °C in laboratory for further experiments.

## DNA extraction

Total bacterial genomic DNA was extracted from fecal samples using QIAamp® Stool Mini Kit (Qiagen, Hilden, Germany) following the manufacturer's protocol. DNA quantity and quality were examined using NanoDrop™ One (Thermo Fisher Scientific, Waltham, MA, USA) according to the manufacturer's instruction.

## Bacterial 16S rRNA genes amplification and sequencing

The V3–V4 hypervariable region of the 16S rRNA gene was amplified using primers 341F (5′-CCTAYGGGRBGCASCAG-3′) and 806R (5′-GGACTACNNGGGTATCTAAT-3′). PCR amplifications were conducted in a total volume of 50 μL mixture containing 6 μL of the template DNA, 25 μL of 2×Taq PCR Master Mix (0.1 U/μL; KHBE, China), 2 μL of each primer (10 μM) and 15 μL ddH$_2$O. The reaction system was then subjected to 1 cycle of initial denaturation at 95 °C for 3 min, followed by 25 cycles at 95 °C for 30 s, annealing at 55 °C for 30 s and extension at 72 °C for 30 s, and a final cycle at 72 °C for 5 min. Stained with SYBR® Safe DNA Gel Stain (Invitrogen, Carlsbad, CA, USA), the PCR products were assessed using electrophoresis in 2% agarose gels and visualized under UV light. The PCR products were purified using the GeneJET (Thermo Fisher Scientific, Waltham, MA, USA).

The sequencing libraries were generated using Ion Plus Fragment Library Kit 48 rxns (Thermo Fisher Scientific, Waltham, MA, USA). DNA concentrations of PCR products were quantified through Qubit and subjected to quality control procedures (*Edgar et al., 2011*; *Haas et al., 2011*; *Martin, 2011*). High-throughput sequencing was performed on a Life Ion S5TM XL (Thermo Fisher Scientific, Waltham, MA, USA) following the manufacturer's instructions.

The data set of our study is available in the Sequencing Read Archive (SRA) on NCBI with accession numbers of SRP149194.

## Sequence processing and data analysis

The original sequencing reads were trimmed using Cutadapt V1.9.1 (http://cutadapt. readthedocs.io/en/stable/) (*Martin, 2011*). Raw reads were obtained after removing barcode and primers. Chimeric sequences were checked and eliminated based on UCHIME Algorithm (http://www.drive5.com/usearch/manual/uchime_algo.html) (*Edgar et al., 2011*) and Gold database (http://drive5.com/uchime/uchime_download.html) in order to generate clean reads.

For all samples, OTUs were generated from clean reads via Uparse v7.0.1001 software (http://drive5.com/uparse/) with a 97% sequence identity cutoff value (*Edgar, 2013*). Using Mothur (*Schloss et al., 2009*), representative sequences of the OTUs which were chosen by the highest frequency of occurrence, were annotated against the SILVA SSUrRNA database (http://www.arb-silva.de/) (*Quast et al., 2013*; *Schloss et al., 2009*; *Wang et al.,*

*2007*) and aligned by MUSCLE (Version 3.8.31) (*Edgar, 2004*) to construct the phylogenetic relationship between different OTUs.

Series of alpha-diversity indices including Observed species, Shannon, Simpson, Chao1, ACE, and Goods coverage were calculated and analyzed in QIIME (Version 1.9.1) (*Caporaso et al., 2010*). The rarefaction curves and rank abundance curves were constructed in R (Version 2.15.3). We applied Wilcoxon rank-sum test to identify discrepancies of gut bacterial diversities between the Amur leopard and North Chinese leopard for each index of alpha-diversity.

Using the QIIME pipeline (Version 1.9.1), beta-diversity was assessed by calculation of Unifrac distances and subsequently visualized by principal component analysis (PCoA). Phylogenetic trees were also built using UPGMA (unweighted pair-group method with arithmetic mean). The principal component analysis (PCA), principal co-ordinate analysis (PCoA) and non-metric multidimensional scaling (NMDS) were calculated using R (version 2.15.3) (*R Core Team, 2013*) so as to evaluate the similarity and discrepancies of bacterial communities among fecal samples based on weighted and unweighted distance matrix. The Analysis of Similarities (ANOSIM) was also used to testify whether there was a significant difference between two groups (*Clarke, 1993*). Beta-diversity was then subjected to Wilcoxon rank-sum test.

## RESULTS

### Overall sequencing data

A total of 1,514,233 raw reads were yielded after high-throughput sequencing of all samples. The data sets were then subjected to quality control procedures which resulted in 1,413,825 clean reads for the 21 samples analyzed. The total number of OTUs was 4,203 at a threshold of 97% sequence identity for all samples.

Alpha-diversity indices including Observed species, Shannon, Simpson, Chao1, ACE and Goods coverage are shown in Table 1. The rarefaction curves showed a pattern of plateau formation (Fig. 1A), indicating that the microbial diversity present in each sample was sufficiently quantified at this sequencing depth. We also analyzed the rank abundance curves to evaluate the abundance and distribution of bacteria taxa (Fig. 1B).

### Bacteria composition and relative abundance

Overall, we identified 28 phyla, 55 classes, 88 orders, 167 families and 344 genera of bacteria in the gut microbiota community from 21 fecal samples of leopards.

For the Amur leopard, *Firmicutes* was the predominant phylum (78.4%) (Fig. 2). *Proteobacteria* (9.6%), *Actinobacteria* (7.6%), *Bacteroidetes* (2.6%) and *Fusobacteria* (1.7%) contributed also to the total composition. At the family level, *Planococcaceae* (30.1%), *Clostridiaceae 1* (17.2%) and *Peptostreptococcaceae* (14.5%) were the top 3 dominant families. At the genus level, *Sporosarcina* was predominant with an abundance of 22.8%, followed by *Clostridium sensu stricto 1* (17.1%) and *Peptoclostridium* (10.2%).

For the North Chinese leopard, *Firmicutes* (68.6%) was the most predominant phylum (Fig. 2), followed by *Actinobacteria* (11.6%), *Fusobacteria* (6.4%), *Proteobacteria* (6.2%) and _*Bacteroidetes* _(6.0%). *Clostridiaceae_1* (19.5%), *Planococcaceae* (16.2%)

**Table 1  Alpha-diversity of fecal microbiota in Amur leopard and North Chinese leopard feces.**

| Sample | Observed species | Shannon | Simpson | Chao1 | ACE | Goods coverage |
|--------|------------------|---------|---------|-------|-----|----------------|
| O1 | 226 | 3.205 | 0.788 | 248.521 | 258.106 | 0.999 |
| O2 | 138 | 2.959 | 0.747 | 173.455 | 190.370 | 0.999 |
| O3 | 85 | 0.683 | 0.146 | 108.400 | 117.380 | 0.999 |
| O4 | 110 | 2.137 | 0.649 | 161.250 | 163.516 | 0.999 |
| O5 | 125 | 1.704 | 0.433 | 144.077 | 157.104 | 0.999 |
| O6 | 324 | 4.183 | 0.867 | 350.757 | 351.019 | 0.999 |
| O7 | 159 | 3.875 | 0.85 | 173.040 | 181.306 | 0.999 |
| O8 | 143 | 3.736 | 0.867 | 172.750 | 174.050 | 0.999 |
| J1 | 282 | 3.489 | 0.666 | 294.364 | 289.486 | 1.000 |
| J2 | 286 | 5.798 | 0.961 | 300.056 | 298.907 | 0.999 |
| J3 | 167 | 4.412 | 0.855 | 194.273 | 195.119 | 0.999 |
| J4 | 213 | 4.086 | 0.868 | 237.474 | 233.225 | 0.999 |
| J5 | 150 | 3.871 | 0.888 | 170.036 | 178.518 | 0.999 |
| J6 | 159 | 3.128 | 0.786 | 202.000 | 203.286 | 0.999 |
| J7 | 161 | 2.563 | 0.722 | 246.550 | 234.682 | 0.998 |
| J8 | 330 | 4.813 | 0.909 | 370.886 | 364.072 | 0.998 |
| J9 | 125 | 3.454 | 0.854 | 151.400 | 160.056 | 0.999 |
| J10 | 132 | 2.946 | 0.791 | 145.500 | 156.117 | 0.999 |
| J11 | 122 | 4.169 | 0.919 | 139.105 | 146.006 | 0.999 |
| J12 | 128 | 3.473 | 0.853 | 308.167 | 201.223 | 0.999 |
| J13 | 122 | 1.395 | 0.333 | 157.286 | 164.553 | 0.999 |

and *Lachnospiraceae* (12.5%) were the three most predominant families. At the genus level, *Clostridium sensu stricto 1* (19.4%), *Sporosarcina* (9.5%) and *Peptoclostridium* (6.1%) constituted the top three genera.

The clustered heatmap showed (Fig. 3A) that the gut bacterial distribution of the Amur leopard and North Chinese leopard were relatively scattered. The unweighted pair-group method with arithmetic means (UPGMA) (Fig. 3B) that display the similarities between sample groups showed a similar result with the clustered heatmap.

## Differences in community composition

The boxplots of alpha diversity were shown in Fig. 4A. Observed species and Shannon diversity indices were tested for the significance of discrepancies between the two sample groups ($p = 0.447$ and 0.210, respectively). The beta-diversity indices were presented in Fig. 4B ($p = 0.441$ and 0.003, respectively), illustrating the discrepancies of gut bacterial communities between different groups. The Analysis of Similarities (ANOSIM) showed the significance level between different groups ($R = 0.02$, $p = 0.335$) (Fig. S1). The heatmap of beta-diversity indices were also plotted in Fig. S2.

Non-metric multidimensional scaling (NMDS) displayed separation in gut microbiota composition of the Amur and North Chinese leopard, and the stress value less than 0.2 could show the discrepancy between samples was 0.110 (Fig. 5A). The PCA plot (Fig. 5B)

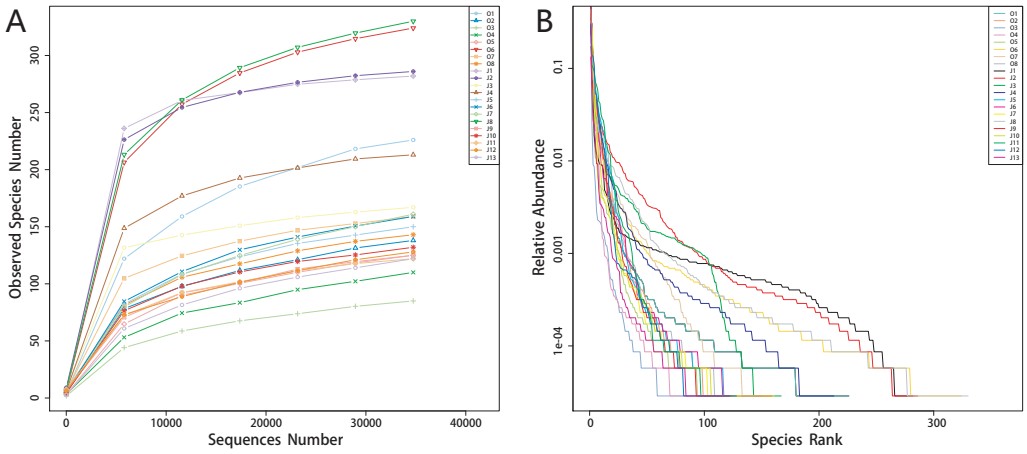

**Figure 1 Rarefaction curves (A) and rank abundance curves (B).** The *x*-axis of rarefaction curves indicates the sequences number selected randomly from samples and the *y*-axis indicates the observed species number (OTUs). The curves in (A) tend to be flat reflect that the sequencing data size is rational. In rank abundance curves (B), the *x*-axis indicates the order number ranked by the OTUs abundance while *y*-axis shows the relative abundance of OTUs. The higher the richness of the species, the larger the span of the curve on the horizontal axis. In the vertical direction, the smoother the curve, the more uniform the species distribution.

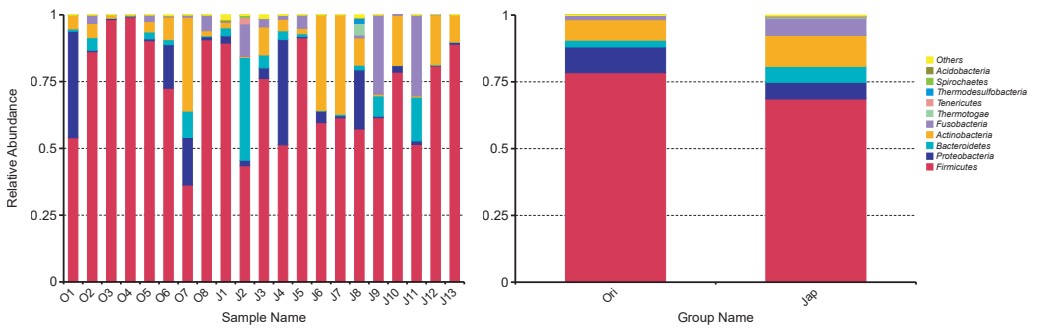

**Figure 2 Fecal bacterial composition of Amur leopard and North Chinese leopard at the phylum level.** The top ten bacterial phyla chosen according to the results of species annotations were ranked by the relative abundance in each sample or group. The *x*-axis and *y*-axis represents the information of samples and relative abundance respectively.

revealed that the samples from the two subspecies were basically clustered together. The main components of the gut microbiota of the Amur leopard and North Chinese leopard were similar with two exceptional samples from the North Chinese leopard. According to the PCoA analysis (Figs. 5C and 5D), the fecal bacterial communities of Amur leopard and North Chinese leopard were relatively scattered in every group. Overall, no significant differences were found between the two sample species according to the results of PCoA, PCA and NMDS.

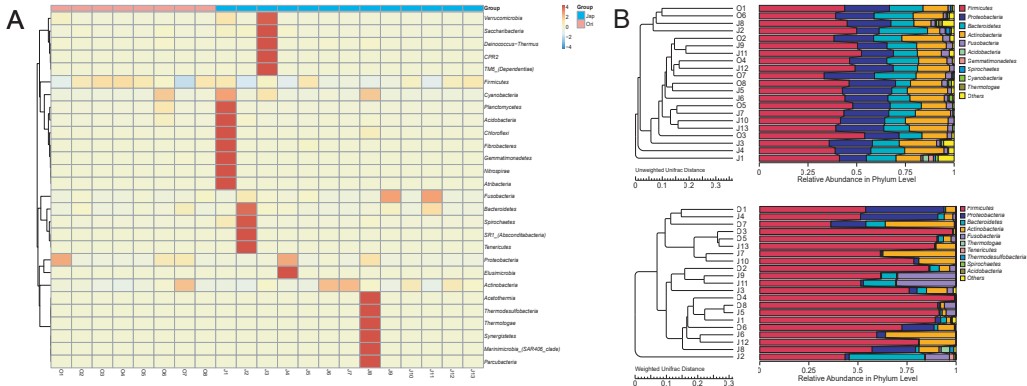

**Figure 3   The heatmap of clustering for species richness (A) and UPGMA clustering trees with relative abuandance in phylum level (B).** The heatmap of clustering for species richness (A) illustrates the bacterial distribution among different fecal samples of leopards. The bacterial phyla were clustered for their relative abundance. The $x$-axis represents each sample, and the $y$-axis represents the relative percentage of each bacterial phyla. The $Z$-value ($-4$ to $4$), displayed by color intensity, is the relative abundance of the sample and the difference in the average relative abundance of all samples divided by the standard deviation of all samples in the classification. In (B), the UPGMA clustering trees were generated with the weighted and unweighted Unifrac distance and then we integrated the trees with the relative abundance of species among all samples in phylum level. The relative abundance of species at phylum changed as the main composition phyla changed based on different Unifrac distance.

## DISCUSSION

With the rapid development of high-throughput sequencing technology, there are mounting studies focusing on the analysis of gut microbiota in different vertebrates. Amur leopards and North Chinese leopards are endangered flagship species facing severe survival predicament result from prey depletion, habitat fragmentation, and anthropogenic disturbances. However, research effort on free-ranging leopards in China, especially for North Chinese leopards are neglected (*Jacobson et al., 2016*). In this study, we characterized and compared the gut microbiota of Amur leopards and North Chinese leopards for the first time using high-throughput sequencing technology. The characterization of their gut microbiota might be able to provide useful information for potential research and help us evaluate the healthy condition of wild leopards in their natural habitat.

Five major bacterial phyla were observed including *Firmicutes*, *Proteobacteria*, *Actinobacteria*, *Bacteroidetes* and *Fusobacteria* both in the Amur leopard and North Chinese leopard samples, which is in accordance with the vertebrate gut microbial diversity described by many other studies (*Deng & Swanson, 2014*; *Ley et al., 2008*; *Ritchie, Steiner & Suchodolski, 2008*). Fecal samples of healthy cats are featured with similar phylum composition with slightly different proportions (*Barry et al., 2012*). Based on our analysis, no significant difference was found in the relative abundance of these five phyla between the samples from the Amur leopard and North Chinese leopard.

*Firmicutes* was the most predominant phylum in both the Amur leopard and North Chinese leopard and showed no significant difference between two groups ($p = 0.210$). Previous researches have reported that *Firmicutes* is the most dominant phylum in feces of

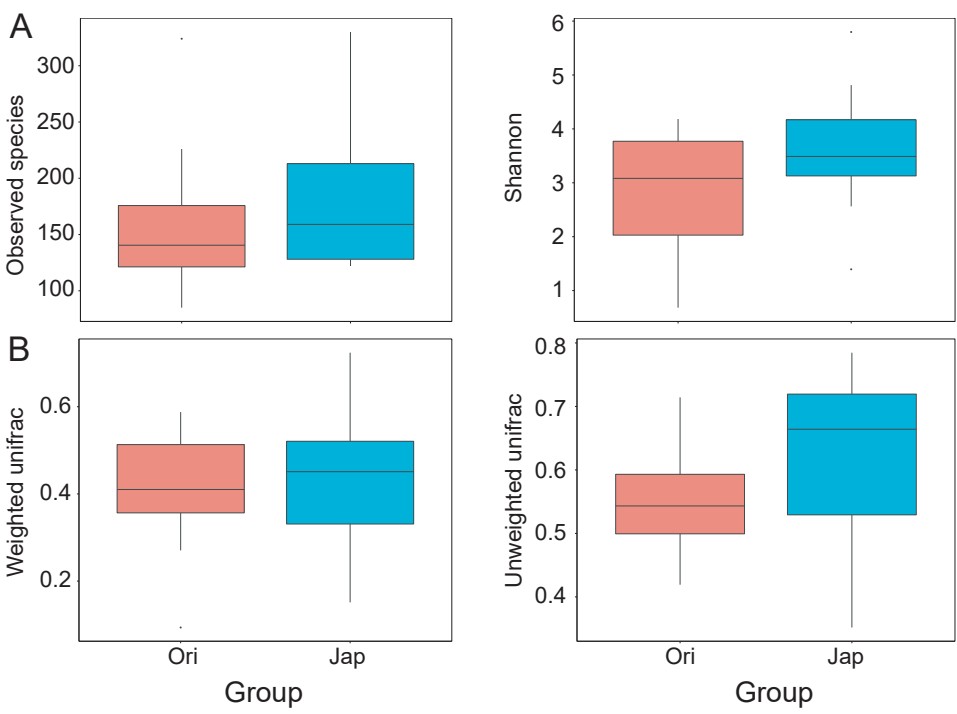

**Figure 4** Comparisons of alpha (observed species and Shannon index) (A), and beta-diversity (with weighted and unweighted Unifrac distance matrix) (B), between Amur leopard and North Chinese leopard fecal samples.

animals (*Garcia-Mazcorro et al., 2012*; *Guan et al., 2017*; *Ritchie et al., 2010*) and humans (*Arumugam et al., 2011*). Same tendency was also found in feline species in the wild such as leopard cats (*Prionailurus bengalensis*) (*An et al., 2017*) and snow leopards (*Panthera uncia*) (*Zhang et al., 2015*). Some studies reported that the body fat storage influences the gut bacterial composition in mice (*Ley et al., 2005*) and humans (*Ley, Peterson & Gordon, 2006*). A significantly greater proportion of *Firmicutes* and a significant reduction of *Bacteroidetes* were observed in obese animals than in lean controls (*Turnbaugh et al., 2006*). The tendency of an increase in *Firmicutes* and a decrease in *Bacteroidetes* was associated with switching to the high fat diet (*Hildebrandt et al., 2009*; *Tremaroli & Bäckhed, 2012*). We detected that the proportion of *Firmicutes* in Amur leopards was relatively greater than in North Chinese leopards, and the proportion of *Bacteroidetes* in Amur leopards was relatively lower which indicated that the weight of Amur leopard should be more heavier. This might relate to the greater body fat storage of Amur leopards compare with North Chinese leopards, since Amur leopards have larger body size and store more fat to withstand severe cold in further north habitat (*Wang et al., 2017*). Unfortunately, the detail information about wild North Chinese leopard is comparatively scarce.

Within the phylum *Firmicutes*, *Zhang et al. (2015)* found that *Lachnospiraceae* was the most diverse family in the feces of snow leopards, which is consistent with a previous report in wolves (*Canis lupus*) (*Zhang & Chen, 2010*). In our results, however, the most

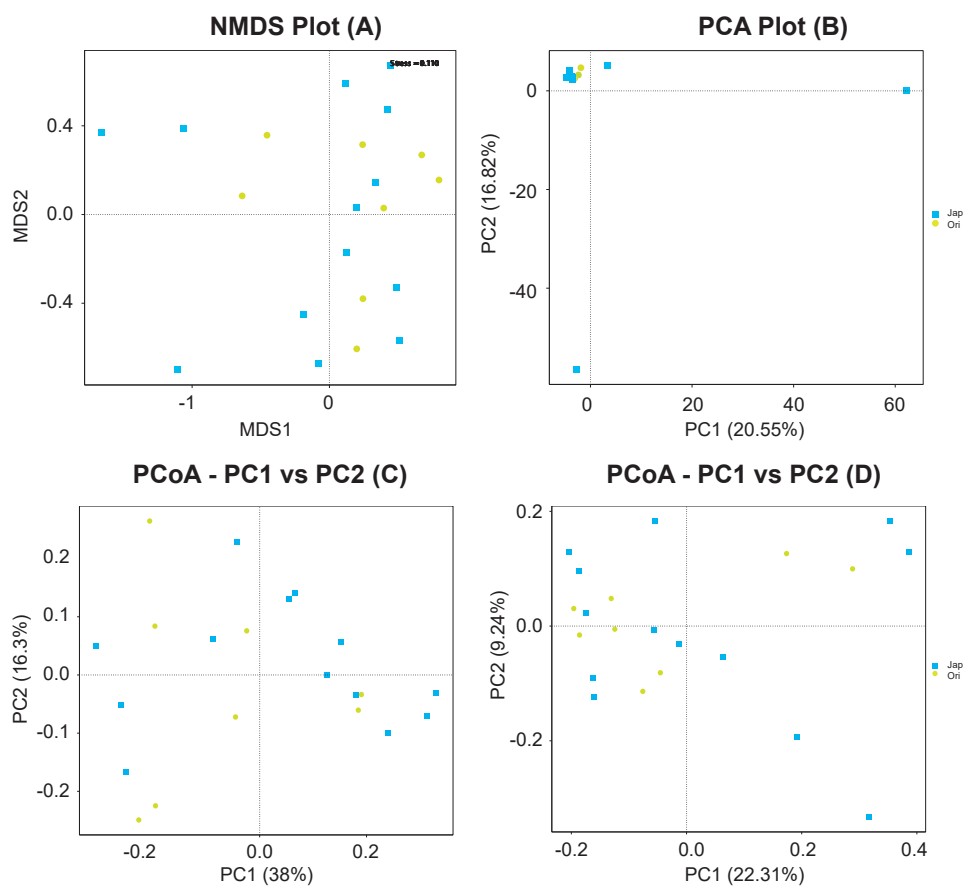

**Figure 5** NMDS (A), PCA (B) and PCoA (C & D) of fecal bacterial community structures of Amur leopard and North Chinese leopard. The yellow points represent Amur leopard and the blue squares represent North Chinese leopard. For PCoA (C) and (D) were analyzed with weighted Unifrac distance and unweighted Unifrac distance respectively. All the points are scattered, which indicates that no significant differences were found between the two subspecies.

diverse family was *Clostridiaceae 1* (19.5% in North Chinese leopard, 17.2% in Amur leopard) within the order *Clostridiales*, and *Lachnospiraceae* constituted a relatively small proportion in our sample set compared to snow leopards and wolves. *Lachnospiraceae* was found in both human and mammal gut microbiota that relates to some diseases like colon cancer (*Meehan & Beiko, 2014*), nonalcoholic fatty liver disease (NAFLD) (*Shen et al., 2017*) and diabetes (*Kameyama & Itoh, 2014*). However, without sufficient support based on other health monitoring methods including blood or apparatus test, the proportion of *Lachnospiraceae* in the gut microbiota could only be a simple referential marker that reflects health condition for wild animals.

Our results also indicated that *Clostridium sensu stricto 1* was a predominant genus in the gut microbiota of leopards. And *Clostridium perfringens* was a common bacterial species for both the Amur leopard and North Chinese leopard. *Lubbs et al. (2009)* reported that the gut microbiota of domestic cats is affected by the protein concentration in diets, particularly, *Clostridium* populations increased as more protein was digested. The presence

of *C. perfringens* was positively associated with protein intake in grizzly bears (*Ursus arctos*) (*Schwab et al., 2011*) and cheetahs (*Acinonyx jubatus*) (*Becker et al., 2014*). To our knowledge, leopards are highly carnivorous and consume mostly protein in their daily diet (*Martins et al., 2011*). We speculate that the high proportions of *Clostridium* populations might reflect the high-protein diet of leopards in our study. Interestingly, *C. perfringens* might be potential pathogenic bacteria that cause diarrhea in dogs (*Canis lupus familiaris*) and cats (*Felis catus*) (*Suchodolski, 2011*). However, *C. perfringens* was also detected in the clinically healthy dogs and house cats (*Handl et al., 2011*; *Queen, Marks & Farver, 2012*). *C. perfringens* should probably be considered as a common commensal in the intestine of healthy feline (*Becker et al., 2014*).

The relationship between gut microbiota and gastrointestinal diseases including inflammatory bowel disease (IBD), chronic enteropathies (CE), and acute diarrhea in Carnivora are well- documented (*Suchodolski, 2016*). For examples, there are increases in the proportions of bacterial genera belonging to *Proteobacteria* and decreases in *Fusobacteria*, *Bacteroidetes*, and *Firmicutes* in canine IBD (*Yasushi et al., 2015*). And an increase of *Enterobacteriaceae* along with decreased proportions of *Bacteroidetes*, *Faecalibacterium* spp. and *Turicibacter* spp. were observed in cats with chronic diarrhea (*Suchodolski et al., 2015*). It indicates that some gut microbiota dysbiosis, which caused by disease processes, can be identified in fecal samples (*Suchodolski, 2016*).

*Proteobacteria* was another phylum in the gut microbiota of the leopards and was not significantly different between two subspecies ($p = 0.804$). *Proteobacteria* was also detected in other feline gut microbiota analysis based on different methods (*Ozaki et al., 2009*; *Ritchie, Steiner & Suchodolski, 2008*). As the most predominant phylum in giant panda (*Ailuropoda melanoleuca*), *Proteobacteria* play crucial role in degrading lignin, which is the main ingredient of bamboo (*Fang et al., 2012*), and in catabolizing complicated compounds in fodder (*Evans et al., 2011*). *Proteobacteria* was the most predominant phylum in the gut microbiota of dogs in the obese groups while in the lean groups was *Firmicutes* (*Park et al., 2015*), and the proportion of *Proteobacteria* was related with inflammatory bowel disorder (IBD) and *Clostridium difficile* infection (*Chang et al., 2008*; *Packey & Sartor, 2009*) as well. For many large mammals in the wild, noninvasive sampling such as collecting feces or hairs are relatively feasible and effective method to obtain information, but the real-time living situation and health condition of some wild species are still unclear, as well as the definite function of different bacteria in host health.

The phylum *Actinobacteria* also contributed to the gut microbiota of the leopards and its proportion was not significantly different between the two subspecies ($p = 0.414$). It was identified to be the most predominant phylum in snow leopards (*Zhang et al., 2015*). In contrast, *Wu et al. (2017)* reported that *Actinobacteria* constituted 0.53% of all gut bacteria in wolves (*Canis lupus*). *Handl et al. (2011)* found that *Actinobacteria* constituted 7.3% of all bacterial sequences in house cats, but was at a low abundance in dogs (1.8%). This result in regard to dogs was in line with an analysis using 454 pyrosequencing (*Middelbos et al., 2010*). There might be a different tendency in the abundance of *Actinobacteria* between feline and canine species, or perhaps the abundance of *Actinobacteria* in the mammalian gut is currently biased, because sequencing methodology without prior %G+C profiling

might underestimate the proportion of high G+C bacteria including *Actinobacteria* (*Harri et al., 2009*).

*Bacteroidetes* was another contributive phylum in the gut microbiota of the leopards and showed no significant difference between groups ($p = 0.707$). *Bacteroidetes* ranks over *Firmicutes* as the most predominant phylum in some cases, as shown in domestic cats and dholes (*Cuon alpinus*) (*Jacobson et al., 2016*; *Tun et al., 2012*). The relative abundance of *Bacteroidetes* varied significantly in different studies (*Handl et al., 2011*; *Ritchie et al., 2010*). Within *Bacteroidetes*, the genus of *Bacteroides* contributed 1.1% and 3.2% to the gut microbiota of the Amur leopard and North Chinese leopard, respectively. *Bacteroides* species were reported that took part closely in the breakdown of complex molecules, such as polysaccharides, also the biotransformation of bile acids (*Lan et al., 2006*; *Reeves, Wang & Salyers, 1997*). Additionally, this crucial genus could inhibit some pathogenic micro-organisms (like *Escherichia coli*, *Klebsiella pneumonia*) with other anaerobic bacteria, which beneficial to host (*Hentges, 1983*; *Van der Waaij, Berghuis-de Vries, 1971 & Lekkerkerk-Van der Wees*). Although the presence of *Bacteroides* in the gut microbiota might be beneficial to the health condition of leopard to some extent, more feces samples should be collected for further investigation with camera trap data and identified individual in the nature reserves to prove the above inferences.

With regard to other phylum *Fusobacteria* detected in the feces of leopards, no significant difference was found between samples from the two subspecies ($p = 1.000$). Research based on pyrosequencing suggested that *Fusobacteria* were less abundant in domestic cats than in domestic dogs (*Garciamazcorro et al., 2011*). This tendency might be analogous to the relationship of gut microbiota observed in raccoon dogs (*Nyctereutes procyonoides*) and leopard cats (*An et al., 2017*).

In summary, although the proportions of the five predominant bacterial phyla are slightly different among the gut bacteria of the Amur leopard and North Chinese leopard, no significant difference was found in phylum composition between the two subspecies. Previous work has shown that the gut microbial community structure of species can vary in different environments (*Clayton et al., 2016*), and respond to dietary alterations, including the amount and type of dietary fiber or other bioactive food components (*Turnbaugh et al., 2008*). The potential prey for the Amur leopard includes Siberian roe deer (*Capreolus pygarus*), sika deer (*Cervus nippon*), wild boar (*Sus scrofa*) and Badger (*Meles meles*) in the Northeast Tiger and Leopard National Park (*Yang et al., 2018*). However, sika deer is not available for North Chinese leopard in Tieqiaoshan Provincial Nature Reserve (*Wu, Wan & Fang, 2004*), which indicates that in addition to the relative different dietary components and living environment, there are other influence factors play crucial roles for the gut microbiota of wild leopard. We speculate that genetic factors might be responsible for the same tendency in gut microbiota composition, after all the classification of this two subspecies leopard is still controversial around the world due to the lack of efficient evidence in molecular biology. At the genus level, however, the bacterial composition for each fecal sample is individualized. Due to the principles of noninvasive sampling, there may be some variables that cannot be measured easily, such as age, sex, real-time healthy condition or dietary shift of wild leopards, which account for the individualized bacterial

microbiota at genus level. More fecal or intestinal samples from wild leopards are required for in-depth analysis of the gut bacterial community. The metabolic pathway of bacterial species should also be taken into account to provide a more comprehensive insight into the functional repertoire of the leopard gut microbiota. Although the implications of changes in gut microbiota for human and other species have been shown in many studies, the implications for wild animal conservation are still limited (*Bahrndorff et al., 2016*). Other studies also suggested that microbiome and fitness of host could be influenced by habitat fragmentation (*Amato et al., 2013*; *Cheng et al., 2015*). For instance, for these rare endangered animals, captivity and reintroduction into the wild are the common methods for their conservation. The gut microbiota of captive animals could be established with excepted nutritional conditions by feeding special diets as the wild individual, and reintroduction would have relative high success rate when captive animals facing uncertain environment. It is necessary to figure out the relationship between habitat fragmentation and gut microbiota, as well as the performance of diverse gut microbiota under different conditions. In general, our study presents the characterization and comparison of the gut microbiota for wild leopards, which might be able to provide a theoretical reference both for free-ranging leopards and ex-situ conservation.

## CONCLUSIONS

We first reported and compared the basic composition and structure of the fecal microbiota between wild Amur leopard and North Chinse leopard. We observed that *Firmicutes*, *Proteobacteria* and *Actinobacteria* were the three most predominant phyla in the gut microbiota of both Amur leopard and North Chinese leopard. Although their living environment and diet are relatively diverse, no significant difference was found in the main composition and structure of the gut microbiota at phylum level. We speculate that the same structure of fecal microbiota might result from genetic factors of leopard, the small sample size or too much variability within the groups. In order to understand the gut microbial ecology of Amur and North Chinese leopards, future research should focus on within-individual variation in microbial community structure, and how gut microbiome structure changes with seasonal shifts in temperature and diet. Furthermore, other methods including functional metagenomics of the gut microbiota, and whole genome sequencing of leopards, integrated with behavioural data from infrared camera traps in the field will be beneficial for leopard conservation.

## ACKNOWLEDGEMENTS

We sincerely thank Tonggang Chen, Shuyun Peng, Zhanzheng Sun, Chunze Tan, Dazhao Song, Shaoping Wan, Shiming Cui, Qiaowen Huang, and Yuelong Chen for field assistance.

### Funding

This study was supported by grants from the National Natural Science Foundation of China (31670537 and 31200410), the National Scientific and Technical Foundation Project of China (2012FY112000) and the Cyrus Tang Foundation. The funders had no role in study design, data collection and analysis, decision to publish, or preparation of the manuscript.

### Grant Disclosures

The following grant information was disclosed by the authors:
National Natural Science Foundation of China: 31670537, 31200410.
National Scientific and Technical Foundation Project of China: 2012FY112000.
Cyrus Tang Foundation.

### Competing Interests

The authors declare there are no competing interests.

### Author Contributions

- Siyu Han and Yu Guan conceived and designed the experiments, performed the experiments, analyzed the data, contributed reagents/materials/analysis tools, prepared figures and/or tables, authored or reviewed drafts of the paper.
- Hailong Dou analyzed the data, contributed reagents/materials/analysis tools.
- Haitao Yang conceived and designed the experiments, performed the experiments, analyzed the data.
- Meng Yao contributed reagents/materials/analysis tools, approved the final draft, revised the manuscript.
- Jianping Ge and Limin Feng conceived and designed the experiments, contributed reagents/materials/analysis tools, approved the final draft, revised the manuscript.

### Field Study Permissions

The following information was supplied relating to field study approvals (i.e., approving body and any reference numbers):

Field experiments were approved by the Forestry Department of Jinlin Province, State Forestry Administration and Forestry Department of Shanxi Province.

### Data Availability

The raw data is at SRA accession number SRP149194.

### Supplemental Information

Supplemental information for this article can be found online at http://dx.doi.org/10.7717/peerj.6684#supplemental-information.

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
