# Peer review of "Comparison of the fecal microbiota of two free-ranging Chinese subspecies of the leopard (Panthera pardus) using high-throughput sequencing"

_PeerJ, doi:10.7717/peerj.6684_

## Round 0.1 · original submission · Major Revisions

This study has some merit, but requires substantial improvement. Two expert reviewers have provided some incredibly detailed comments that should help you in this regard. I particularly agree with the reviewers that the title needs amending to more accurately reflect the study.

From my own reading of the manuscript, I would identify two major areas for improvement. First of all, the introduction is quite weak and does not adequately set up the rationale for the study. It begins with a very general section on the importance of gut microbiota, and the some discussion on the taxonomy of leopards, but at no point doe sit adequately make the case for why studying the gut microbiota of these two closely related species might be important, or yield more insight into how closely they share nice space etc. Overall the introduction therefore felt quite disjointed. Second, the discussion is overly long and contains a lot of superfluous information. You spend too much time talking about phylum level classifications and why certain phyla might be important, when actually *genus* is still too coarse a scale to be able to understand functional differences performed by these bacteria.

If the two groups you have investigated don't differ in overall gut community structure, then that might be interesting from a management perspective, and might have implications for understanding how different their diets are. You don't really address this until the last paragraph starting at Line 339. Don't feel you have to fill the discussion with text to justify the paper. Use this space to interpret the results you have.

Finally, although you mention in the discussion how difficult it is to retrieve these samples, I would like you to acknowledge that your sample size is very small even for understanding group level differences in gut microbiome structure (i.e. before looking at individual level traits such as age, sex etc).

Additional Comments
Line 38: do you mean bacterial phylum composition, or OTU composition? Even if you’re using UNIFRAC as a distance measure, these analyses are not testing at the phylum level

Line 41 has laid “the” foundation;

Line 41: suggest you change to “investigations of the health, dietary preferences and physiological..”

Line 67: if you are going to mention the range of leopards perhaps cite a relevant reference such as Jacobson et al 2016 PeerJ

Jacobson AP, Gerngross P, Lemeris Jr. JR, Schoonover RF, Anco C, Breitenmoser-Würsten C, Durant SM, Farhadinia MS, Henschel P, Kamler JF, Laguardia A, Rostro-García S, Stein AB, Dollar L. (2016) Leopard (Panthera pardus) status, distribution, and the research efforts across its range. PeerJ 4:e1974 https://doi.org/10.7717/peerj.1974

Line 85/86: why is the relationship between the Amur and North Chinese Leopard ‘perplexing’? And who will studying the gut microbiota help resolve this? You have an opportunity here to frame the introduction in a much clearer way

Line 176: perhaps use ‘quantified’ instead of comprehended

Line 211: remove comma after ‘ANOSIM’

Line 222: should this be “from the North” rather than “froorth”?

Line 226: why use three different tests for differences?

Line 229: why ‘remarkably’ higher? You provide no quantitative estimates of differences

Line 237: inconsistent referencing of ‘Re M et al ‘, and again line 257

Line 243: why is it imperative for conservation to study the gut microbiota?

Line 281 onwards: this whole paragraph spends a lot of manuscript space to talk about the link between Clostridium and diet, saying that ‘To [your] knowledge, leopards are highly carnivorous”. You could be a lot more concise here and simply say you found a lot of Clostridia and that this genus is associated with carnivorous diets. I don’t think comparisons to the gut microbiome of newborn infants is particularly helpful here.

Line 299: phrases like ‘pathomechanisms” are unclear and unhelpful jargon

Line 300-307: This whole paragraph should be deleted. Talking about functions bacteria perform when classified at the phylum level is pointless, and you add no new information here

Line 308-332 Same here as for the above. These phylum level characterisations are not very helpful

Line 355: You mean unmeasured variables, not unidentified.

Reviewer 1 ·

Basic reporting

General recommendations:
♣ Please revise all in-text reference formatting and adapt to meet the author’s instructions provided by the journal.
E.g. L46: (Nicholson, Holmes et al. 2012) is the wrong formatting. Should be (Nicholson et al., 2012)
♣ Please revise the full reference list formatting and adapt to meet the author’s instructions provided by the journal.
E.g.  JB, C., V. P, H. H, W. T, H. BM, A.-G. GA, T. DA, L. HT, T. BV and M. VV (2016). "Captivity humanizes the primate microbiome." Proceedings of the National Academy of Sciences of the United States of America 113 (37): 10376. > Surnames should not be abbreviated.
E.g.  Katherine R Amato, Carl J Yeoman, Angela Kent, Nicoletta Righini, Franck Carbonero, Alejandro Estrada, H Rex, Gaskins, Rebecca M Stumpf, Suleyman Yildirim, Manolito Torralba, Marcus Gillis, Brenda A Wilson, Karen E Nelson, Bryan A White and S. R. Leigh (2013). "Habitat degradation impacts black howler monkey (Alouatta pigra) gastrointestinal microbiomes." Isme Journal 7 (7): 1344-1353. > Abbreviate author names, use correct Journal abbreviations, …
L623: remove abstract from reference!
TIP: Use a decent referencing program with a plugin for your writing program (e.g. Mendeley, EndNote, Reference Manager)
♣ Please consider an alternative title that more precisely describes the study. “Analysis of gut microbiota for” is too general nowadays. Try to refer to the type of analysis (diversity analyses) and two different subspecies of leopards. The latter is important given the emphasis throughout the manuscript on similarities and differences between both subspecies.
E.g. Fecal microbial diversity composition in free-ranging North Chinese and Amur leopards as revealed by Ion Torrent 16S rRNA gene sequencing
E.g. Phylogenetic diversity analysis of fecal microbial communities in free-ranging North Chinese and Amur leopards as determined by ...
♣ Capitalize ‘north Chinese leopard’ and change throughout manuscript to North Chinese leopard
♣ Introduction: Clarify more precisely the current findings on the relationship between conservation (and associated important factors of it) and gut microbial composition.
♣ Figures: increase quality of all figures and rewrite figure legends more detailed
♣ Figure 3b: there is a marked difference in relative abundance at Phylum level resulting from unweighted or weighted Unifrac distances (e.g. Proteobacteria). Please elaborate on that in your results or reconsider the use of either one or the other, depending upon your research objective.


Detailed manuscript review:
INTRODUCTION
♣ L45: Change ‘The characterization of gut bacterial community’ into ‘Characterization of gut bacterial communities’
♣ L47: remove following part ‘the gut bacterial community structures of a species can vary substantially when the environment changes from wild to captivity (ref)’ and refer to this aspect at the end of this alinea. Thus, first state some general well-known aspects of host-microbiota associations (e.g. role in diet, host health, and physiology, act as a barrier against pathogens, …) and then refer to association microbiota with wild/captive settings.
♣ L47: Question: Please elaborate on findings from reference provided in L47 (JB et al., 2016). Given you work on samples from free-ranging animals and state throughout the manuscript that the information gathered could be useful for conservation strategies, it is necessary to adapt your introduction more towards this aspect. Moreover, this then leads into alinea two starting at L56.
♣ L56: ‘The gut microbiota analysis’ … Remove ‘the’ because there is no such thing as “The gut microbiota analysis” and change into more specified wording e.g. gut microbial diversity analyses or deciphering gut microbial composition and function, …
♣ L59-L61: unclear how exactly these aspects (host habitat, diet and gut microbiota) are related. Write less generally the findings from Amato et al., 2015
♣ L61-L63: Likewise, state the findings from Gomez et al., 2015 less general! Replace “Increased anthropogenic pressure could also be distinguished via the composition of the gut bacterial community” > how was the relationship between the increase of anthropogenic pressure and gut bacterial community composition (did this increase, change, decrease, …?)
♣ L63: Besides. And disease-associated bacteria. Replace by ‘Additionally, it has been shown that the detection of pathogenic bacteria …’.
♣ L65-66: Reformulate by emphasizing that knowledge on gut microbial composition can contribute to conservation strategies by its impact on different factors important for conservation (habit fragmentation, pathogen load, …)
♣ L69: Remove in the past century
♣ L75-L76: rephrase to ‘Once patrolling from Northeast China to southernmost portions of the Russian Far East and the Korean peninsula, …’
♣ L79: rephrase to ‘The North Chinese leopard originally distributed North and Central China. However, an accurate distribution area and population size still remain unclear …’
♣ L83-86: rephrase to clarify following sentence ‘included the North Chinese leopard in Amur leopard on account of the obscure biogeographical barrier’. Why are they subspecies in the leopard clade?
♣ L89: Question: Why are leopards elusive in their feeding behavior? Is ‘elusive’ the correct term to use in this context?
♣ L89: rephrase and correct the following sentence ‘it is acknowledged that the gut microbiota relates to adaptive evolution to a high purine and fat diet of carnivores’. Please clarify and specify what aspect of the gut microbiota: composition, richness, functional pathways, … has been shown to adapt following an evolution to a high purine and fat diet? Consider including additional references, this has been shown by different authors.
♣ L95: change ‘gut bacterial communities’ to ‘fecal bacterial communities’
♣ L96: change ‘hyper-variable’ to ‘hypervariable’
♣ L97-L98: remove this part of the sentence ‘and discussed the implications … these endangered species.’ This has not been discussed in-depth in the discussion section. Suggestion to state that this study aims to provide the first benchmark of gut microbial diversity in leopards that potentially can contribute to further conservation research.

MATERIAL AND METHODS
♣ L101: Rephrase to ‘Opportunistic fecal sampling occurred in the period from December 2016 to March 2017 in the two distribution areas from the leopards.’
♣ L155: Change ‘could help preserve …’ to the more academic wording ‘contributed to the preservation of gut microbes in the fecal samples’.
♣ L155: What is considered a “fresh” fecal sample by the team? This is of utmost importance when analyzing afterward the fecal bacterial composition. (cfr. General recommendations)
♣ L116: I assume storage at -80℃ refers to laboratory storage. What were in-field storage and transport conditions? Were samples collected on ice, dry ice?
♣ L118: Total bacterial genomic DNA
♣ L118-L119: Please add methodologies for the check of DNA quantity and quality after DNA extraction procedures.
♣ L121: hypervariable
♣ L125: ‘and15μL ddH2O’ > insert a space
♣ L133: Question: What quality control procedures were applied?
♣ L134: change ‘Life Ion S5TMXL’ to correct brand name ‘Life Ion S5TM XL’
♣ L139-L140: Remove ‘Chimeric sequences can contribute to … excluded (ref)’. This should not be mentioned in the Material and Methods section. Just state immediately how chimeric sequences were checked for and eliminated.
♣ L142-L143: Formulate more appropriately. Chimeric sequences were detected by the UCHIME algorithm and the chimeras were removed using the ChimeraSlayer utility (references).
♣ L144: Rephrase to ‘For all samples, OTUs were generated from clean reads via Uparse v…’
♣ L149: This sentence is unclear and not necessary ‘all taxonomic information was obtained to form the composition of the gut bacterial community’. Later on in the manuscript, the authors state that many OTUs remained unclassified so not all taxonomic information was obtained. Please remove the sentence.
♣ L145-L150: Please rephrase more correctly this methodology. E.g. Using Mothur (Schloss et al., 2009), sequences were annotated against the SILVA SSUrRNA database (http://www.arb-silva.de) (reference) and aligned by MUSCLE (ref).
♣ L153: Rephrase to ‘calculated and analyzed in QIIME’
♣ L154: The Rarefaction curves and rank abundance curves were constructed in R (Version 2.15.3).
♣ L155 – L157: I do not agree with the statistical methods applied. Please reconsider these. One should start by identifying the nature of the distribution of values of the indices. Knowing the nature of the data in microbial ecology and the small sample size, it seems highly unlikely to me that these data are distributed normally and homogeneous. Moreover, why use then both parametric and non-parametric test?!
♣ L157: correct name of the test is Wilcoxon Test (wilcox.test is the command line in R)
♣ L158: Do not capitalize beta-diversity
♣ L158-L159: Please rephrase more correctly this methodology. E.g. Using QIIME pipeline (version), beta-diversity was assessed by calculation of Unifrac distances and subsequently visualized by principal component analysis (PCoA). Phylogenetic trees were also built using UPGMA (unweighted pair group method with arithmetic mean)
♣ L159: UPGMA = unweighted pair group method with arithmetic mean
♣ L159-L163: I would recommend to choose one visualization analyses and not include all three of them i.e. PCA, PCoA, and NMDS! Moreover, the results and discussion do not mention potential different outcomes of these methodologies.
♣ L163-L166: Question: Why do you choose these statistical methods? Why use parametric and non-parametric tests for paired group testing and then ANOSIM (parametric method) to test this same data.

RESULTS
♣ L167: Question: Can you explain an LDA score of 4?
♣ L174: were > are shown
♣ L174: Do not capitalize rarefaction curves
♣ L174: What do you mean by “the stable values”? To my knowledge, rarefaction curves do not need to reach a certain value but show a pattern of plateau formation
♣ L176: use an alternative verb for ‘comprehended’
♣ L176: Do not capitalize rank abundance curves
♣ L178-179: Please move this sentence to Materials and Methods
♣ L181: change ‘we identified’ to ‘X number of OTUs (%) could be taxonomically assigned to 28 phyla, 55 classes, … .’. I would like to see the percentage of OTUs that could be taxonomically assigned. Since most of the results are partitioned into Amur leopard vs North Chinese leopard, please report also the number of sequences recovered for each sample set separately. I would like to know if there was a difference between that because this will also impact your discussion on the percentage of taxonomic groups present.
♣ L185: were also contributed to > Other phyla included Proteobacteria (x%), …
♣ L186 and L188: please remove underscore and use proper bacterial nomenclature (please check also elsewhere in the manuscript)
♣ L196: Remove sentence (should be in Materials and Methods)
♣ L197-L198: Please mention these sample codes for both groups in Materials and Methods already.
♣ L200-201: Remove sentence ‘This analysis was generated using both weighted Unifrac distance and unweighted Unifrac distance respectively.’ This again refers to Material and Methods, avoid redundant information.
♣ L203-L205: Remove the first sentence since it encompasses an explanation about a boxplot, one should know what it represents. No need to cover that in the manuscript.
♣ L206-L213: Rephrase this whole section and report the results observed from the methodologies instead of rewriting the methodology. This is of utmost importance for a result section.
♣ L222: froorth??
♣ L224: Question: What does that mean ‘the stress value was 0.110’? Is that relevant to mention in the manuscript result section??
♣ L227-L228: Remove that first sentence, it is a methodology description. F.ex. Fron LEfSE it was shown that the relative abundance of Bacillaceae is remarkably higher …

DISCUSSION
♣ L23: for > in
♣ L233-L238: Remove these sentences. This is a repetition from the introduction and not directly relevant to the outcome of your study since you do not link any metadata except for host species to your data.
♣ L238-L240: Question: Why are these studies important for the conservation of endangered species?! What relationships have been found between factors of conservation and gut microbiota?! How can you apply knowledge on gut microbial composition to species conservation?
♣ L243-245: Question: Why is this analysis imperative for research in wild leopard conservation? (cfr previous comment, elaborate more specifically on the application of this knowledge to species conservation)
♣ L243-245: Question: Why would we consider these analyses necessary if obtaining samples is actually not easy and hardly feasible. With this statement ‘…tremendous hardship in obtaining non-invasive samples or intestinal part of endangered leopard has impeded the characterization of their gut microbiota’, you undermine your own study and this contradicts with the outcome of your rarefaction and rank-abundance curves.
♣ L245-L247: Question: Finding multiple unclassified genera, is this due to small sample size or methodology or reference database?! Knowing current databases already include sequences from many different animals (including different large carnivores), how many novel sequences (%) would you then expect to find in leopards?!
♣ L250: Please rephrase to ‘which is in accordance with the vertebrate gut microbial diversity described by many other studies’ and add reference Ley et al., 2008 (https://www.nature.com/articles/nrmicro1978)
♣ L253-L254: Please clarify: ‘no significant difference in phyla composition? Relative abundance?
♣ L261: insert space for ‘some studies reported that’
♣ L262: ’is associated’ instead of ‘was associated’
♣ L263: Please correct to ‘the tendency of an increase in Firmicutes and a decrease in Bacteroidetes…’
♣ L266: futher > further
♣ L265-L267: It is already a well-known given in the field that the proportion of Firmicutes/Bacteroidetes might reflect differences in dietary pattern. Please elaborate on that and be as specific as possible in your discussion. As it is stated now, it is too vague knowing the body of literature out there.
♣ L268-L273: Remove this section out of the discussion unless you actually discuss these findings in the light of other studies. As it is written know, it is a reporting of results, which should occur in the result section.
♣ L280: ‘..a relatively small proportion compared to this family in the gut microbiota of wild leopards’.
♣ L280: Clarify the phrase above, it is not clear as written now. Did you mean Lachnospiraceae constituted a relatively small proportion in your sample set compared to snow leopards and wolves instead of wild leopards?! Also include reference there.
♣ L280: Question: Any idea on why you observe a difference between leopards and snow leopards and wolves?
♣ L283: the gut microbiota was affected by > gut microbial composition is affected by
♣ L288: Question: Elaborate on this and add additional references that underpin the impact of protein content in the diet on Clostridium populations in carnivores [e.g. Becker et al., 2014 (in cheetahs), Schwab et al. 2011 (in grizzly bears), Zentek et al., 2003 (in dogs and cats)]
♣ L289-L292: Remove sentence ‘In addition, research on …’ as it does not fully fit into the discussion. Such parameters (e.g. sepsis, inflammation, antibiotics) have not been measured in your study.
♣ L297: Include more references on the finding of Clostridium perfringens in strict carnivores (non-domestic such as cats and dogs). Cfr previously mentioned references
♣ L294-L299: Suggestion to include some background findings on the presence of gastrointestinal diseases in wild or captive leopards.
♣ L302: include a space before your reference
♣ L317: Rephrase so it is more clear that the G+C content leads to a bias because of the implication of a high G+C content on methodologies
♣ L322-L323: Please remove sentence on storage conditions unless you put your results in light of those findings from Maukonen et al., 2012. Moreover, if you consider your storage conditions to be suboptimal for assessment for microbial diversity, why only discuss in the light of Bacteroidetes?
♣ L328-L331: Please be more specific on what nutrients and pathogens. All gut bacteria somehow contribute to the uptake of nutrients for the host and competition with pathogens (natural colonization barrier)
♣ L331-L332: Please discuss your findings in the light of the total observed microbial composition in relation to leopard health, not only Bacteroides
♣ L344-L345: Please report critically: how much change in bacterial composition do you expect to see that could be attributed to nutritional differences between both host species?! Maybe this difference cannot be observed because it does not distinguishable from normal variation in their gut microbial diversity or sample size is too small or …?
♣ L348: Knowing the differences in their diet, would it have been advisable to also analyze the feces for prey composition?!
♣ L353: Please explain “internal factors”
♣ L354: Please remove either free-ranging or wild, they mean the same.
♣ L360: cfr previous remark, if you question your sampling storage conditions, your whole study is questioned. Nowadays, good guidelines exist to storage fresh fecal samples for next-gen sequencing diversity studies. If proper storage is assured, it should be of a lesser impact then the multitude of other variables influencing microbial gut composition.

Experimental design

This is an original primary research within the Aims and Scope of the journal. Although the research question is well defined, the manuscript lacks in the discussion section a critical reflection of these results in the light of conservation of leopards. Please be more precise, too many statements are too general about gut microbial diversity.

Following changes are suggested to improve the technical standard. Take special care to the rationale of performed statistic tests and visualization analyses.

♣ Materials and Methods: Provide more information on fecal sample collection > what is a fresh sample, how much time after defecation was it collected? E.g. How long was the sample outside in the environment? Meteorological conditions during sampling? Include as well how the samples were collected: what (sterile?!) recipients were used?
♣ Material and Methods: I do not agree with the statistical analyses using parametric as well as non-parametric tests. The nature of these data and the small sample size should point towards a non-homogenous and non-normal distribution of the data. Test with assumptions of homogeneity and normal distribution should therefore not be used.
♣ Material and Methods: Quantitative (unweighted Unifrac) and qualitative (weighted Unifrac) beta-diversity measures lead to different insights into factors that structure microbial community composition. Assess to what extent unweighted or weighted Unifrac should be preferred for this study.
♣ Results: Please improve quality of figures 3a and 3b.
♣ Results: Reporting of results should be restricted to results, not include again Material and Methods.
♣ Results L214-L226: cfr. Material and Methods. Reconsider the value of using three different visualization analyses.

Validity of the findings

♣ Discussion: When reporting from other studies, please use correct verb tense. (see also detailed manuscript review)
♣ Discussion: When reporting results from microbial diversity studies, please specify whether these findings were also obtained by culture-independent high-throughput sequencing. Some of the differences observed between microbial diversity studies might be mainly attributed to different technologies used!
♣ Conclusion: L361-L370 Please rewrite conclusions to include actual well stated results of this study, linked to the original research question.

Although we encourage microbial diversity studies in endangered host species, the authors should put their findings more in relation to specific aspects of conservation. Too often it remains to speculative or vague without an actual foundation of why this information should be relevant to collect from endangered species.

Additional comments

Thank you for your submission.
Please go carefully over the manuscript review comments ranging from important general comments to detailed grammatical comments.

Focus on how to increase the validity of findings by choosing the appropriate statistical test and visualization techniques (instead of performing different incomparable tests alongside each other), by increasing quality of figures and writing concise but informative figure legends.

Report results strictly in result section, and rephrase wording in discussion section so that the results are actually discussed and not repeated in a 'reporting' format.

Although the text generally flows and is written in proper scientific English, please consider remaining grammatical errors. Please take care of referencing (and the reference list) in the correct format, following the author's guidelines of the journal.

Reviewer 2 ·

Basic reporting

The manuscript “Analysis of gut microbiota for free-ranging leopards (Panthera pardus) in China using high-throughput sequencing” by Han et al. describes a comparison of the gut microbiota of two subspecies of leopards using faecal samples as a surrogate. The gut microbiome of carnivores is still not well understood today and the current study adds to that limited knowledge. Although I consider the study to be mostly solid, I was struggling to understand some analyses performed. The manuscript would benefit greatly from improved clarity of the text and figure quality. I would like to see the work published after a major revision.
Introduction, 3 pages:
Most of the first two paragraphs of the introduction (lines 45 -- 66) deal with topics not relevant to the current study. I would like to see the introduction and discussion much more focussed: a bit of background on microbiome in wild carnivores, why the authors expect differences between the two populations, etc.

Methods, 3.5 pages:
Line 145: “The species sequences were annotated” the authors are probably referring to the representative sequences of the OTUs. Also, how were the representative sequences chosen? E.g. most abundant or longest sequence per cluster?
150: “The sequences were then aligned using MUSCLE software (Version 3.8.31)(Edgar 2004) to construct the phylogenetic relationship between different OTUs.” How was the phylogenetic tree inferred?


Res, 3.5 pages
Description of relative abundances of phyla, families and genera in the text is quite lengthy and I feel a table would be more appropriate. I think it is important the authors provide their annotated OTU table as well as summary tables on the different taxonomic ranks so that readers can investigate taxa of interest.
Figures: There are 7 figures in total although the PDF generated states 8. There is redundant information in the figures, for example Figure 2 and Figure 3b illustrate relative phylum abundance. Figure 2 and 3 could be combined without losing information. Figure 5 could go into supplementary and Figure 7 could be excluded completely.
I also feel that figure quality should be improved. Figures blur when zooming in. I cannot tell if this is an issue with the original figures or the PDF.

Figure 1:
A colour scheme that makes it easy to distinguish the two groups would be helpful. Also, the legend describes what rarefaction and rank abundance curves are and not what the figure shows. This is also true for some of the other figures.

Figure 2:
Legend says “bacterial species” but shown are phyla.

Figure 3:
The title is misleading/not informative: “The dendrogram (a) and UPMGA clustering trees (b).” Shown is a heatmap and a bar chart of relative phyla abundance with the dendrogram of the clustering as added information.
a) I do not understand the colour coding of the relative abundances. According to the legend the values range from -4 to 4. This means the values were scaled. How?
b) I do not understand why phylum relative abundance is different between weighted and unweighted Unifrac. Using different distances can affect the clustering but why the abundances? The legend says “The unweighted Unifrac distance and weighted Unifrac distance were used to calculate the overall percentages of relative abundance among all samples at phylum level.” Distances are calculated based on the relative abundances, not the other way round. The legends of the two subplots differ in size.

Figure 5:
As no structure is revealed by the analysis, I feel this figure could go into supplementary information.

Figure 6:
Please change the colour scheme to accommodate for red-green blind readers.

Figure 7:
I question the relevance of this figure. Only three taxa were different. Do we really need a bar chart and a cladogram to show this?

Discussion: 6.5 pages.
The discussion is extensive and, in my opinion, not always relevant to the results. Often bacterial taxa from the current study are mentioned along with other studies that also found these taxa. The studies cited are then compared much like in a literature review.
Also, some parts are better suited for the results section, e.g. lines 268—273 describe results.
I suggest the authors concentrate on the main taxa and where they have been previously been described in other carnivores and on why one might expect differences in the microbiomes of the two subspecies (like they did in lines 339 – 353).

Conclusions
The conclusions do not refer to the findings of the paper.

Experimental design

There is no original field study approval document provided, only a text describing why the authors do not provide the original, i.e. that it is in Chinese. The editor needs to decide if this is acceptable.

There are different numbers of individuals in the two groups (8 vs. 13) which could be problematic for the statistical analyses. However, as no big claims about significant differences are made I do not consider this an issue. The authors should, however, discuss this issue as a potential explanation for the lack of statistically significant differences.

Validity of the findings

The main findings are descriptions of the predominant taxa in the two subspecies of leopards and a lack of statistically significant differences between the two groups (with the exception of the three taxa identified in the LDA Effect Size analysis). I consider these findings valid based on the current experimental design.

Additional comments

• The title does not indicate that two subspecies are being compared. Suggestion: “Comparison of the fecal microbiota of two free-ranging Chinese subspecies of the leopard (Panthera pardus) using high-throughput sequencing”.
• I was not able to access the raw sequence data in SRA under SRP149194.
• As many other papers in the field faecal samples are used to study the gut microbiome. Faecal samples can be used as a proxy but I think this should be mentioned in the text. For example the first sentence of the abstract reads: ”The analysis of gut microbiota provides a non-invasive approach to understand the complex interactions between host species and their intestinal bacterial community.” This is not correct. Obtaining gut samples is invasive, collecting faecal samples and using them as a proxy for the gut is non-invasive.
• References are not formatted correctly. Please refer to author instructions for the correct format.
• In-text citations of multiple references should be separated by semicolons instead of commas.

---

## Round 0.2 · Minor Revisions

Your manuscript has been reassessed by two original reviewers. Collectively, we all agree that the manuscript has improved, but there are still some issues that require attention. Both reviewers have highlighted changes that will benefit the manuscript, and so I am returning a decision of Minor Revisions.

I still find the reasoning behind why this study is important for conservation hard to follow. The introduction lists numerous reasons why the gut microbiota might be a good indication of host health in the introduction, but don't really explore this in the discussion, or hw it might be used for conservaton. Reviewer 1 has highlighted this also. I think discussion space is best used to make practical recommendations about how the results of this study could effect changes in conservation strategies, as this will elevate the impact of the manuscript.

In addition, Line 428 - 437 (Tracked Changes Version): I have an issue with the attribution of function to bacterial phyla here, as I think this can cause confusion. Bacterial strains responsible for certain metabolic functions may belong to the same phylum, but this gives the impression you can measure function, and functional differences at the phylum level.

Reviewer 1 ·

Basic reporting

note: numbering of lines applies to the reviewed manuscript without track-changes

INTRODUCTION
L56-57: Rephrase sentence "North Chinese leopard originally distributed North and Central China while lost as much as 98% of their historic range." to clarify. While? -> but
L62: North Chinses -> North Chinese
L64: supported -> supporting
L65: Amur Leopards -> Amur leopards; north Chinese -> North Chinese
L69-L70: Rephrase sentence "Gut microbial diversity analyses based on leopard fecal samples should also be taken into account conservation efforts" to clarify. Should conservation efforts take into account microbial diversity analyses or vice versa?! I believe you refer to the first option but that does not come from this sentence.
L77: remove"richness" as it is encompassed in the term "diversity"
L78: characterized with -> under

METHODS
L121: traces like footprints -> footprint traces
L121: each transects -> each transect
L122: intervals -> interval
L123: was contributed to -> contributed to
L176: to detecting -> to detect

RESULTS
L193-194: were also contributed to -> contributed also to
L198: remove 'also'
L209: The Boxplots -> The boxplots
L218: remove 'accurately' and 'here'
L221-L224: Rephrase full sentence starting with "According to PcoA..." to clarify. What do you want to say exactly? What do you mean by scattered within groups? Due to the use of different visual representation of data, the key message tends to get a bit lost.

DISCUSSION
L228-230: Please rewrite the sentence, with attention for use and tense of verbs. Grammar in that sentence is not correct. "Amur and North Chinese leopards are endangered flagship species..."
L230: were -> are
L259: north Chinese. Since -> North Chinese, since
L261: North Chinses -> North Chinese
L271: include -> including
L275: north Chinese -> North Chinese
L275: (Lubbs et al., 2009) -> include ref in a sentence without brackets when used as a subject, so Lubbs et al. (2009) suggest ... (check author guidelines for in-text use of references)
L276: domestic cat -> domestic cats
L277: the C. perfringens -> C. perfringens
L288: were -> are
L293: Please rephrase, it is not sure that these diseases cause the dysbiosis or are the result of a dysbiosis.
L296: add a verb for the second part of the sentence
L299: plays -> play
L299-L301: Adds unnecessary information to the discussion. The role of Proteobacteria for lignin digestion is not relevant for a strict carnivore.
L301: remove "plus", not a scientific phrasing
L303: which phylum do you refer to with "this phylum"? Proteobacteria? Moreover, this sentence is too long and confuses since C. difficile belongs again to Firmicutes. Please rephrase more clearly and shorten.
L305: samplings -> sampling
L306: remove 'certain' because this is too vague
L307: refine your statement from "wild species" to "some" wild species, because for some species there is already a lot of information available.
L308: the role of bacteria in gut health/function or host health? Please remove "gut microbiology"
L309: remove 'was'
L310-L311: Remove redundant sentence " the relative abundance of..." because this is just a bulk sentence with no novel information or specific information
L327: Please rephrase and avoid the verb 'related to'. Simply by just mentioning the functions of Bacteroides species.
L329-L331: Please rephrase end of this sentence. Grammar is not correct. "anaerobic bacteria which beneficial the host"?
L333-L334: Rephrase "above interferences" and I suggest you reconsider the statement that more fecal samples are required. Consider more the type of information you can get from a 16S diversity study compared to more functional analyses or metagenomic analyses! You don't need necessarily more samples to find out more about the microbial community but a different type of analyses. Especially when you reference work that points out functions of bacterial species.
L349-L352: I don't think your analyses can support this statement. Please consider carefully. Diversity analyses have been mainly discussed at the phylum level, the dietary impact might manifest at lower taxonomic levels and/or only be assessed on functional analyses. Moreover, specific dietary regimes and nutrient intake for the animals included in the study have not been recorded.
L356-L358: Many of the reported variables (sex e.g.) are not unmeasurable! They have just not been recorded for the animals included in your study and or require a more defined set of descriptors (e.g. biomarkers for health, age ranges based on certain anatomical or physical or physiological characteristics, ...)
L358: Is it the number of samples that is the limiting factor? Consider the analyses performed. Will you gather necessarily more useful and "in-depth" information with just more samples?
L361: Please remove "in the hope of acquiring" ... It is evident that scientific studies have as an outcome "to acquire more knowledge by gathering new and/or more data".
L361-L364: Please consider this statement carefully and check the most recent literature. High-throughput sequencing technology can already provide much more in-depth information than provided in this study and the technology in itself should not be scrutinized as such.
L365: Please consider the use of "normal" as it puts forward the question highly discussed "what is a normal host-associated microbiota?!"
L369: remove "using high-throughput sequencing" and "after analyzing 21 wild feces samples'; this is redundant info
L373-L374: specify the taxonomic level for this statement
L378: remove "into the prospective research direction"
L378: include -> including

Experimental design

Comments on experimental design are incorporated in the 'Basic Reporting' section.

Validity of the findings

While the introduction and discussion section have improved, the authors still lack to incorporate a critical reflection of the results in the light of leopard conservation. Which aspects of leopard conservation can benefit from this study? How can leopard conservation be addressed via the study of the gut microbiota? e.g. Likely functional analyses and dietary analyses will provide more inside. But the introduction also highlights the impact of habitat fragmentation on gut microbial communities of species living in the wild but this is not mentioned again in the discussion. Wouldn't studies considering both gut microbial diversity and ecological and anthropogenic factors benefit in-situ conservation? Which type of conservation (ex-situ or in-situ) can benefit from microbial analyses? Diet-gut microbiota relations (as suggested in the discussion by mentioning prey type) might be of bigger importance in captive settings (ex-situ conservation) where a diet is an important management tool.

Relative abundance levels at the phylum level between leopards and other species have been discussed, but the authors should finalize this manuscript with a reflection of these data in the light of conservation.

Additional comments

Thank you for your re-submission and addressing in detail most of the comments and suggestions from the first review. The manuscript has improved. Please go carefully over this second manuscript review which includes some remaining grammatical comments and critical remarks about some of the statements made.
Please consider the advice in section 3 (Validity of findings). The discussion section could be shorter in the comparative description about relative abundances of phyla and include more interpretation of these results in light of conservation of leopards.

Reviewer 2 ·

Basic reporting

The manuscript “Comparison of the fecal microbiota of two free-ranging Chinese subspecies of the leopard (Panthera pardus) using high-throughput sequencing” by Han et al. is a resubmission of a manuscript I reviewed previously. It describes a comparison of the gut microbiota of two subspecies of leopards using faecal samples as a surrogate. The gut microbiome of carnivores is still not well understood today and the current study adds to that limited knowledge. The study has some merit and the resubmitted version is already much improved in terms of quality of the figures and clarity of the text. Grammatical issues remain in the text but do not impact clarity. I would like to see the work published after my remaining issues with the figures are clarified.
Introduction:
The introduction is now better tailored to the results of the study.
Methods:
Lines 157—162: What version of the Silva database was used? How were the phylogenetic trees inferred after aligning the sequences with Muscle?
Results:
I still think it would be beneficial if the authors provided their annotated OTU table as supplementary information so that readers can investigate taxa of interest.
The results section feels disconnected from the figures that it should be describing.

Figures:
Figure quality was improved overall.
Figure 1:
As previously mentioned, the legend describes what rarefaction and rank abundance curves are. This is not relevant. The figure legend should briefly summarize the findings. This is also true for some of the other figures.
Figure 3:
a) The figure or the legend should state that the values shown in the heatmap are z-scores. The authors explained this in their rebuttal letter but choose to omit this information from the figure.
b) I still do not understand why phylum relative abundance is different between weighted and unweighted Unifrac. Please explain this to me and the readers and double check your data. As no OTU table is provided I am unable to interrogate the data myself.

Figure 4:
The last sentence “The x-axis and the y-axis indicate groups and the indices of different diversity respectively. “ is not required.

Figure 5:
The red-green colour scheme is still not ideal.
Discussion:
The discussion now is more relevant to the results of the paper.

Conclusions
I am fine with the conclusion except from the following statement. “We infer that the same tendency of fecal microbiota might mainly result from genetic factors of leopard.” I do not think this can be inferred from the data as such. You may be able to speculate this but it is more likely the lack of significant differences is caused by the small sample size and too much variability within the groups.

Experimental design

The design is typical for studies of fecal samples from the wild. Limitations such as freshness and the limited number of samples are inherent and have been addressed.

Validity of the findings

The findings are observational. As I do not have access to the data I am unable to verify them.

---

## Round 0.3 · Minor Revisions

Thank you for making the requested changes to your manuscript. I have reassessed the paper, and not the following areas where clarity of writing could be improved.

The following numbers refer to the tracked changes MS Word version of the manuscript.

Line 116: “a total of 13 fecal samples”
Line 300: carnivora -> Carnivora
Line 326: wolfs -> wolves
Line 357-359: “Known evidence elucidates that…” -> “Previous work has shown that the gut microbial community structure of species CAN vary…”
Line 372: “there may be some variables THAT cannot be measured easily…”
Line 376: “also be TAKEN into account to provide a more comprehensive insight into THE FUNCTIONAL REPERTOIRE OF THE LEOPARD GUT MICROBIOTA…”
Line 377-381: Delete sentence beginning “Although the rapid development…”
Line 381-: Rewrite sentence “Although differences in gut microbiome structure have been linked to differences in host health in humans and other species, the consequences of changes in gut microbiome structure for leopards and their conservation remain unresolved.
Line 384/385: It’s not clear what “inhabitation fragment” means. Do you mean influenced by habitat?
Line 387/388: The phrase “treated with excepted nutritional conditions” doesn’t make sense. Are you talking about priming the gut microbiome of captive animals prior to reintroduction by giving them specialised diets?
Line 390: “inhabitation fragments” -> “habitat fragments” ?
Line 402: tendency -> structure
Line 404: Suggest change to “…to understand the gut microbial ecology of Amur and Northern Chinese leopards, future research should focus on within-individual variation in microbial community structure, and how gut microbiome structure changes with seasonal shifts in temperature and/or diet. Furthermore, other methods including functional metagenomics of the gut microbiome, and whole genome sequencing of leopards, integrated with behavioural data from infrared camera traps in the field will be beneficial for leopard conservation. ”

---

## Round 0.4 · accepted · Accept

Many thanks for making the required changes on the manuscript. I am now happy to accept the paper in its revised form.

#